# A Paper-Based Electrochemical Sensor Based on PtNP/COF_TFPB−DHzDS_@rGO for Sensitive Detection of Furazolidone

**DOI:** 10.3390/bios12100904

**Published:** 2022-10-21

**Authors:** Rongfang Chen, Xia Peng, Yonghai Song, Yan Du

**Affiliations:** National Engineering Research Center for Carbohydrate Synthesis, Key Lab of Fluorine and Silicon for Energy Materials and Chemistry of Ministry of Education, College of Chemistry and Chemical Engineering, Jiangxi Normal University, Nanchang 330022, China

**Keywords:** paper-based electrode, reduced graphene oxide, covalent organic framework, furazolidone, electrochemical sensor

## Abstract

Herein, a paper-based electrochemical sensor based on PtNP/COF_TFPB−DHzDS_@rGO was developed for the sensitive detection of furazolidone. A cluster-like covalent organic framework (COF_TFPB−DHzDS_) was successfully grown on the surface of amino-functional reduced graphene oxide (rGO-NH_2_) to avoid serious self-aggregation, which was further loaded with platinum nanoparticles (PtNPs) with high catalytic activity as nanozyme to obtain PtNP/COF_TFPB−DHzDS_@rGO nanocomposites. The morphology of PtNP/COF_TFPB−DHzDS_@rGO nanocomposites was characterized, and the results showed that the smooth rGO surface became extremely rough after the modification of COF_TFPB−DHzDS_. Meanwhile, ultra-small PtNPs with sizes of around 1 nm were precisely anchored on COF_TFPB−DHzDS_ to maintain their excellent catalytic activity. The conventional electrodes were used to detect furazolidone and showed a detection limit as low as 5 nM and a linear range from 15 nM to 110 μM. In contrast, the detection limit for the paper-based electrode was 0.23 μM, and the linear range was 0.69–110 μM. The results showed that the paper-based electrode can be used to detect furazolidone. This sensor is a potential candidate for the detection of furazolidone residue in human serum and fish samples.

## 1. Introduction

Antibiotics are widely used in medicine and aquaculture. However, the harm caused by the use of antibiotics is irreversible. In addition, antibiotic drugs are harmful to the body’s immune system, the liver, and the kidneys. The abuse of antibiotics is extraordinarily serious, and about 700,000 people die directly or indirectly every year. Furazolidone is a nitro-containing antibiotic drug, which has been widely used in aquaculture and veterinary medicine to minimize the acute effects of Escherichia coli, Shigella, Salmonella, and other infections [1,2,3]. Owing to the accumulation of furazolidone through the food chain, it is detrimental to the human body’s immune system and even causes mutation and carcinogenesis [4,5]. Therefore, it is urgent to develop a rapid method to realize the ultrasensitive detection of furazolidone.

Covalent organic frameworks (COFs) are crystalline organic porous polymers assembled by covalent bonding of C, N, O, and other light elements, whose structure can be regulated by the periodic arrangement of organic structural units [6,7]. COFs have stable networks and open channels, which are extraordinarily conducive to the transport of electrolyte ions and have great potential in various applications. It has been reported that some COFs or COF-based sensors have better performance [8,9,10]. However, COFs are highly conjugated, and the π–π interaction between layers would lead to a decrease in specific surface area and poor electrical conductivity.

The development of COF composite materials to enhance electrical conductivity and specific surface area has been proven to be a feasible route. The sp^2^ hybridization of reduced graphene oxide (rGO) and its extremely thin atomic thickness enable rGO to be used in many fields [11,12,13,14]. The rGO-based composites are beneficial to improve the overall conductivity of the composites. COFs can be covalently connected with rGO-NH_2_ to improve the electrical conductivity of the composites. In addition, COFs are also ideal nanospacers that can reduce the stacking of adjacent rGO nanosheets and improve ion transport [15,16,17]. Furthermore, the presence of numerous chelating sites in the COF structure could be employed to immobilize metal nanoparticles through coordination effects. The precise anchoring of nanoparticles in the ordered holes of COFs can also avoid self-aggregation [18]. 

Although chromatography, immunoassay, and fluorometry exhibit high sensitivity and accuracy in the detection of antibiotic residues in water and meat products, their limitations are the complex and time-consuming sample-handling procedures, and the corresponding expensive instruments and equipment. The electrochemical method is not only simple to operate but also has high sensitivity, accuracy, and portability of equipment, which effectively overcomes the above shortcomings [19]. A paper-based analytical device was developed by Whitesides for the first time. Electrochemical paper-based analytical devices usually consist of a three-electrode setup integrated into a paper substrate, offering several benefits, such as reduced consumption of reagents and samples, portability, low cost, and availability of the raw material. They are widely used in various fields [20,21,22,23,24].

In this work, rGO was converted into rGO-NH_2_ through amino functionalization, which was employed as the substrate material. Then, COF_TFPB−DHzDS_ with abundant chelating sites were prepared through the amine-aldehyde condensation reaction of 2, 5-bis (3-(ethyl thiol) propoxy) p-benzoyl hydrazine and 1,3,5-tri(p-formylphenyl) benzene. The -NH_2_ is beneficial for the COF_TFPB−DHzDS_ to grow on rGO-NH_2_ uniformly to form COF_TFPB−DHzDS_@rGO. COF_TFPB−DHzDS_ has abundant N and S atoms; therefore, Pt^4+^ can be precisely doped in its surface and internal structure through coordination and adsorption. Lastly, PtNP/COF_TFPB−DHzDS_@rGO was prepared by using an in situ reduction method. The PtNP/COF_TFPB−DHzDS_@rGO exposes tremendous catalytic activity. A paper-based electrochemical device based on PtNP/COF_TFPB−DHzDS_@rGO composites on flexible peeled graphite paper for the detection of furazolidone was proposed. It can effectively avoid the time-consuming polishing and cleaning work of the conventional electrode, and the test results showed good performance. The device was expected to be an all-in-one electrochemical platform for the detection of antibiotics. This work provides a reference for the design and fabrication of integrated paper-based electrodes and their application in electrochemical sensing. 

## 2. Experimental

### 2.1. Materials and Reagents

2,5-bis(3-(ethylthiol) propoxy) p-benzoyl hydrazine (DHzDS) and 1,3,5-tri (p-formylphenyl) benzene (TFPB) were purchased from Jilin Yanshen Technology Co., Ltd. (Beijing, China). Glacial acetic acid, anhydrous acetonitrile, tetrahydrofuran, ethanol, methanol, 1,4-dioxane, N, N-dimethylformamide, hydrochloric acid, and mesitylene were purchased from Inokay Co., Ltd. (Beijing, China) The rGO, sodium borohydride, tetranitrotetrafluoroborate diazonium salt, tetrabutylammonium tetrafluoroborate, potassium chloroplatinate, zinc powder, ammonium chloride, furazolidone, and other chemicals were purchased from Aladdin’s Reagent Network (Shanghai, China). Flexible graphite paper was purchased from Jinglong Special Carbon Technology Co., Ltd. (Beijing, China). Glassy carbon electrode (GCE) was purchased from Chenhua (Beijing, China) Instrument Co., Ltd. All reagents are analytically pure and do not require further purification when used. Additionally, 0.2 M a phosphate-buffered solution was prepared from 0.2 M sodium dihydrogen phosphate and 0.2 M disodium hydrogen phosphate solution in different proportions. The ultra-pure water was purified by using a Millipore-Q System (ρ ≥ 18.2 MΩ cm).

### 2.2. Instruments

Transmission electron microscopy (TEM) was performed using A JEM-2010 (HR) instrument. Scanning electron microscopy (SEM) images were collected by using a HITACHI S-3400N instrument, and the breakdown voltage was set to 15 kV. X-ray powder diffraction (XRD) analysis was performed using a D/Max 2500 V/PC instrument with Cu Kα radiation from 2° to 35° at a scanning rate of 1°/min. N_2_ adsorption/desorption isotherm tests were carried out with Autosorb-iQ (Quantachrome) under 77 K. All electrochemical tests were performed at the CHI760D (Shanghai, China) Electrochemical Workstation in a conventional three-electrode system (platinum wire as an auxiliary electrode, saturated calomel (SCE) as a reference electrode, and different modified GCEs as working electrodes for routine testing, or homemade paper-based electrodes including graphite-like foam electrodes as working electrodes and counter-electrodes and Ag/AgCl as the reference electrode). Cyclic voltammetry (CVs) and differential pulse voltammetry (DPV) tests were carried out in 0.2 M static N_2_ phosphate-buffered solutions. The control frequency range of electrochemical impedance (EIS) was 0.01–105 Hz, and the amplitude was 5 mV, using 5.0 mM Fe (CN)_6_^3−/4−^ as the signal probe. The data plotted for the calibration curve are the averages of the five experiments, and the length of the error bar indicates the magnitude of the relative deviation.

### 2.3. Synthesis of rGO−NH_2_, COF_TFPB−DHzDS_, COF_TFPB−DHzDS_@rGO, and PtNP/COF_TFPB−DHzDS_@rGO

Preparation of rGO-NH_2_: First, 12 mg rGO was dissolved in a 10 mL anhydrous acetonitrile solution, and the solution was ultrasonic for 1 h. Then, 24 mg tetrafluoroborate diazonium salt and 329 mg tetrabutyltetrafluoroborate ammonium were added, and all the reagents were dissolved and stirred in the dark for 20 h. The solution was dried through centrifugation, and then 10 mg precipitate was added into a 40 mL ethanol–water solution (ethanol: water = 3:2). Next, 104 mg zinc powder, 2 mg ammonium chloride, and 2.3 mL glacial acetic acid were added to the solution, and the mixed solution was heated in 60 °C water bath for 3 h. After cooling down to room temperature, the rGO-NH_2_ was obtained by alternate cleaning with ethanol and water and drying in a 60 °C oven.

Preparation of COF_TFPB−DHzDS_: Briefly, 23 mg TFPB and 8.1 mg DHzDS were dissolved in a 3 mL mixed solution composed of mesitylene and 1,4-dioxane in a ratio of 3:1. Then, 600 μL of 6 M acetic acid was added to the solution and reacted at 120 °C for 3 days. Finally, it was cleaned 5 times with THF and dried for 6 h in a 60° coven.

Preparation of COF_TFPB−DHzDS_@rGO: The steps were similar to COF_TFPB−DHzDS_. Briefly, 1.62 mg TFPB and 4.6 mg DHzDS were dissolved in a 3 mL mixed solution, then 10 mg rGO-NH_2_ was added, and the same concentration and volume of acetic acid solution were added after the mixture was evenly mixed. The follow-up procedure was the same as the preceding procedure.

Preparation of PtNP/COF_TFPB−DHzDS_@rGO: The dried 4 mg COF_TFPB−DHzDS_@rGO was dispersed into a 3 mL methanol solution, and then 2 mL 0.012 mM potassium chloride platinate solution was added and stirred at room temperature for 12 h. Next, 20 μL 0.25 M sodium borohydride solution was added, stirred for 6 h, and washed with ethanol and methylene chloride until the upper solution became colorless and transparent. Finally, the product was dried in a freeze-dryer for 4 h. The preparation process of PtNP/COF_TFPB−DHzDS_@rGO materials is shown in Figure 1.

### 2.4. Preparation COF_T__FPB−DHzDS_ and PtNP/CO_FTFPB−DHzDS_@rGO/GCE

First, 2 mg COF_TFPB−DHzDS_ or PtNP/COF_TFPB−DHzDS_@rGO was dispersed in a 1 mL DMF solution and dissolved via sonication. Then, 10 μL suspensions were dropped onto polished GCE and dried naturally for subsequent experiments.

### 2.5. Preparation of Paper-Based Electrodes (ePADs)

First, the commercial carbon paper was cut into long strips of 3 mm in width and 3 cm in length by using a regular paper knife. Then, two sides of a white cardboard sheet 2 cm in width and 3 cm in length were painted with white nail polish. Next, three long carbon paper strips were pasted onto a side of the white cardboard with an interval of about 0.5 cm. The middle sections of the long carbon paper strips (about one-third of the long strips) were painted with white nail polish. The bottom sections of the long carbon paper strips were peeled off using acrylic transparent tape to obtain a new surface with graphite-like foam as electrodes (working electrode, reference electrode, and counter electrode). Then, the reference electrode was coated with a layer of conductive silver powder, and HCl was added dropwise to form AgCl/Ag. (Figure 2).

### 2.6. Preparation of PtNP/COF_TFPB−DHzDS_@rGO/ePAD

The procedures were similar to PtNP/COF_TFPB−DHzDS_@rGO/GCE. Briefly, 2 mg PtNP/COF_TFPB−DHzDS_@ rGO was dispersed in a 1 mL DMF solution, dissolved through sonication, then 10 μL suspensions were dropped on the lower 1/3 of the working electrodes of the paper-based electrodes. Finally, the electrode surface was dried under a tungsten lamp for 5 min.

## 3. Results and Discussion

### 3.1. Characterization of COF_TFPB−DHzDs_

The SEM image (Appendix A) showed that COF_TFPB−DHzDs_ was filamentous and could stack into clusters under the interaction of the Van der Waals force. FTIR spectrum (Appendix A) demonstrated the successful synthesis of COF_TFPB−DHzDs_. The stretching vibration peak of C=N at 1620 cm^−1^ appeared in COF_TFPB−DHzDs_, proving the formation of an imine bond and confirming that COF_TFPB−DHzDs_ was formed through an amine-aldehyde condensation reaction between monomer TFPB and DHzDS [25]. The XRD pattern (Appendix A) showed strong diffraction peaks at 4.77° and 26.3°, which belong to (100) and (001) crystal planes, respectively, indicating that COF_TFPB−DHzDs_ has a good crystal structure [26,27]. According to the N_2_ adsorption and desorption isotherms (Appendix A) of COF_TFPB−DHzDs_, the specific surface area of COF_TFPB−DHzDs_ was about 153.76 m^2^ g^−1^, and the average pore size was about 1.9 nm. Its specific surface area was large, suggesting it could be used as an excellent support material [28]. 

### 3.2. Characterization of PtNP/COF_TFPB−DHzDS_@rGO

The SEM (Figure 1a) and TEM (Figure 1d) images showed that rGO was a large two-dimensional slice with slight folds [29,30]. After the amino functionalization, COF_TFPB−DHzDS_ could uniformly grow on the surface of rGO (Figure 1b). Compared with rGO (Figure 1a, d), the surface of COF_TFPB−DHzDS_@rGO became rough. Further investigation of the surface morphology using TEM showed that the thickness of COF_TFPB−DHzDS_@rGO increased, compared with the rGO, confirming the synthesis of COF_TFPB−DHzDS_@rGO (Figure 1e). Figure 1c shows the SEM image of PtNP/COF_TFPB−DHzDS_@rGO, whose morphology was not significantly different from that of COF_TFPB−DHzDS_@rGO. Figure 1f shows that a large number of PtNPs with diameters of about 1 nm were uniformly loaded on COF_TFPB−DHzDS_@ rGO (Appendix A). The formation of small PtNPs can be ascribed to the fact that Pt^4+^ uniformly adsorbed on the interlaminar structure or channel of COF_TFPB−DHzDS_ during the synthesis process because COF_TFPB−DHzDS_ had double chelating sites of N or S atoms [31].

The FTIR spectrum (Figure 2a) showed that rGO was basically a straight line without obvious adsorption peaks. After the amino functionalization, it could be observed that the stretching vibration peak of -NH_2_ appeared in the spectrum of around 3453 cm^−1^, which strongly proved that rGO was successfully functionalized. From the spectrum of COF_TFPB−DHzDS_@rGO, it could be found that, in addition to the stretching vibration peak of -NH_2_ at about 3442 cm^−1^, there was also a vibration peak at 1630 cm^−1^, which was attributed to C=N vibration peaks. The C=N vibration peaks were observed after the aldehyde condensation (Figure 2a), indicating that COF_TFPB−DHzDS_ was grown on the surface of rGO-NH_2_ successfully [32]. In the X-ray photoelectron spectrum (Figure 2b) of PtNP/COF_TFPB−DHzDS_@rGO, it could be observed that the material was composed of elements such as C, N, O, and Pt. As shown in Figure 2c, 72.45 eV and 75.80 eV corresponded to the peaks of Pt^0^4f_7/2_ and Pt^0^4f_5/2_, respectively, which proved that the material contained PtNPs. In Figure 2d, a strong peak mainly appeared at 284.01 eV in the C 1s fine spectrum, which belonged to C−C or C=C, while the three strong peaks appearing at 397.98 eV, 399.06 eV, and 399.88 eV in Figure 3e corresponded to =NH, −NH− and −N=N−, respectively [33,34,35]. In the XPS fine spectrum of O 1s (Figure 2f), the peak at 530.70 eV corresponded to C−O, while the peak at 532.60 eV corresponded to C=O. In summary, taking advantage of the large specific surface area and good electrical conductivity of rGO, and the porous framework structure and double chelation sites of COF_TFPB−DHzDS_, Pt^4+^ was in situ reduced through a chemical reduction method, and a large amount of small-sized PtNPs were uniformly loaded on COF_TFPB−DHzDS_@rGO. This work provides a new reference for the construction of composite materials.

### 3.3. Electrochemical Behaviors of PtNP/COF_TFPB−DHzDS_@rGO/GCE

Figure 3a shows the CV curves of GCE, COF_TFPB−DHzDS_/GCE, and PtNP/COF_TFPB−DHzDS_@rGO/GCE in a 0.1 M KCl solution containing 5 mM [Fe (CN)_6_]^3−/4−^. The bare GCE had a pair of reversible redox peaks with a peak-to-peak potential difference (ΔEp) of 85 mV. After the modification with COF_TFPB−DHzDS_, the peak current was obviously reduced, and ΔEp was increased to 118 mV, indicating that COF_TFPB−DHzDS_ inhibited the electron transfer. The PtNP/COF_TFPB−DHzDS_@rGO-modified electrode had a larger peak current and smaller ΔEp (about 98 mV). This suggested that PtNPs and rGO enhanced the reversibility of the reaction and significantly improved its electrochemical performance. This result is mainly attributed to the excellent electrical conductivity of rGO and PtNPs. Figure 3b shows the EIS of COF (curve a), COF_TFPB−DHzDS_@rGO (curve b), and PtNP/COF_TFPB−DHzDS_@rGO. The results indicated that PtNP/COF_TFPB−DHzDS_@rGO has the smallest charge transfer resistance, similar to GCE. The CVs of PtNP/COF_TFPB−DHzDS_@rGO/GCE in a 0.1 M N_2_-statured phosphate-buffered solution (pH = 7.0) with 10 µM furazolidone showed an obvious reduction of furazolidone and an indistinct oxidation peak. According to a previous report [36], the redox mechanism is speculated to be that the nitrogroup contained in the structure of furazolidone is reduced under the synergistic catalysis of Pt and rGO. With the increase in the scanning rate, the peak current density of furazolidone increased (Figure 3c), and a good linear relationship was presented (Figure 3d), indicating that the reaction process was a typical surface control process [37,38].

### 3.4. Optimization of the Experimental Conditions

The amount of COF_TFPB−DHzDS_ growing on the rGO surface, the amount of PtNPs immobilized on COF_TFPB−DHzDS_@rGO, the pH value of electrolyte solution, and the concentration of PtNP/COF_TFPB−DHzDS_@rGO dispersion had significant effects on the performance of the constructed electrochemical sensor, and accordingly, these conditions were optimized in order to realize the sensitive detection of furazolidone (Figure 3e–h).

Figure 3e indicates that with the amount of COF_TFPB−DHzDS_ on rGO, the current responses gradually increased. When the amount exceeded 10 μM, the current responses decreased because COF_TFPB−DHzDS_ seriously accumulated. The amount of COF_TFPB−DHzDS_@rGO was maintained at 3 mg, and the current response of PtNP/COF_TFPB−DHzDS_@rGO prepared at different concentrations of potassium chloroplatinate to furazolidone was investigated. It could be seen intuitively from Figure 3f that, when the concentration of potassium chloroplatinate was 12 μM, the resulting sensor had the optimum current response to furazolidone. When the concentration of potassium chloroplatinate was further increased to 16 μM, the peak current density of furazolidone decreased instead. The reason might be that the ion concentration was too high, thus leading to the larger size of the formed PtNPs with decreased catalytic ability. The detection of furazolidone in a 0.2 M phosphate-buffered solution with different pH levels by using PtNP/COF_TFPB−DHzDS_@rGO/GCE was studied, and the results are shown in Figure 3g. It could be seen that the peak current density value reached the maximum at pH = 7.0. Therefore, a phosphate-buffered solution with pH = 7.0 was used as the optimal electrolyte solution for the detection of furazolidone by using PtNP/COF_TFPB−DHzDS_@rGO/GCE. Finally, the amount of PtNP/COF_TFPB−DHzDS_@rGO modified on GCE was optimized, and the results are shown in Figure 3h. As the concentration of PtNP/COF_TFPB−DHzDS_@rGO increased, the catalytic current of furazolidone gradually increased and reached the maximum value at 2 mg mL^−1^. When the concentration of PtNP/COF_TFPB−DHzDS_@rGO exceeded 2 mg mL^−1^, the PtNP/COF_TFPB−DHzDS_@rGO accumulated on the surface of GCE, and the modified layer became thicker, which hindered the mass transfer, resulting in a decrease in the peak current density value. Therefore, 2 mg mL^−1^ of PtNP/COF_TFPB−DHzDS_@rGO was selected as the optimal modification concentration.

### 3.5. Electrochemical Sensors Based on PtNP/COF_TFPB−DHzDS_@rGO for Furazolidone

The DPV curves of GCE, COF_TFPB−DHzDS_/GCE, and PtNP/COF_TFPB−DHzDS_@rGO/GCE in a 0.2 M N_2_-saturated phosphate-buffered solution (pH = 7.0) with 30 μM furazolidone showed that the PtNP/COF_TFPB−DHzDS_@rGO/GCE exhibited excellent furazolidone reduction performance with a large reduction peak current and a positive reduction potential, thus providing a sensitive method for furazolidone determination (Figure 4a). The highly sensitive response of PtNP/COF_TFPB−DHzDS_@rGO/GCE to furazolidone was attributed to the high catalytic activity of PtNP/COF_TFPB−DHzDS_@rGO and the strong interaction between the PtNP/COF_TFPB−DHzDS_@rGO and furazolidone [39,40]. The high electrical conductivity of PtNP/COF_TFPB−DHzDS_@rGO accelerated the electron transfer to furazolidone during the redox process, and the rich aromatic system in COF_TFPB−DHzDS_ facilitated π–π stacking interactions with conjugated molecules such as furazolidone. Furthermore, the large surface area of the PtNP/COF_TFPB−DHzDS_@rGO provided abundant sites for analyte binding and reduction. PtNPs as nanozymes catalyzed the reduction of furazolidone, and rGO was beneficial to improve the conductivity. Under the synergistic effect, the peak current became larger. Due to the existence of PtNPs, the overpotential of the reaction was reduced, making furazolidone easier to reduce, so the more positive the potential was, the more the peak potential shifted to the right.

Next, the current responses of PtNP/COF_TFPB−DHzDS_@rGO/GCE in furazolidone solutions with different concentrations were measured. Figure 4b shows the response of DPV to furazolidone with different concentrations. The results showed that the peak current response increased as the concentration of furazolidone increased, which maintained a linear relationship over a wide range (15.0 nM−110 μM). Figure 4c shows the corresponding linear relationship between the peak current density and the concentration of furazolidone. Each value was the average of the resulting values that were repeated five times. The linear equation was *j* = –84.6511*c*–3.2922 (R^2^ = 0.99), where *j* and *c* were the peak current density and the concentration of furazolidone solution, respectively. The detection limit of PtNP/COF_TFPB−DHzDS_@rGO/GCE for furazolidone was 5.0 nM. 

Compared with other electrochemical sensors, this sensor also had some advantages (as shown in Table 1) [41,42,43,44,45,46,47,48,49]. Compared with NiCo_2_O_4_@C/GCE and NST/GCE, PtNP/COF_TFPB−DHzDS_@rGO/GCE had a lower detection limit, while compared with NST/GCE and rGO/GCE, the constructed furazolidone sensor had a wider detection range. Then, the selectivity of PtNP/COF_TFPB−DHzDS_@rGO/GCE was tested. As shown in Figure 4d, the PtNP/COF_TFPB−DHzDS_@rGO/GCE showed good selectivity in the presence of potentially interfering substances. The repeatability and stability of PtNP/COF_TFPB−DHzDS_@rGO/GCE were also investigated. Six different PtNP/COF_TFPB−DHzDS_@rGO/GCE electrodes were prepared and used to detect 100 μM furazolidone with a relative standard deviation of 0.93%, which proved that the prepared sensor had good repeatability (Figure 4e). Subsequently, the same PtNP/COF_TFPB−DHzDS_@rGO/GCE was tested seven times after different storing times, and the current signal only decreased by 11.80% after 30 days, indicating that PtNP/COF_TFPB−DHzDS_@rGO/GCE also had good stability (Figure 4f).

Figure 5a,b shows the CV and EIS curves of ePAD and PtNP/COF_TFPB−DHzDS_@rGO/ePAD in a 0.1 M KCl solution containing 5 mM [Fe (CN)_6_]^3−/4−^. The bare ePAD had a pair of reversible redox peaks. After the modification of the target material, the peak current slightly decreased, and the impedance slightly increased. These experimental results were similar to those of GCE, which proved that the paper-based electrode was successfully prepared. The current responses of PtNP/COF_TFPB−DHzDS_@rGO/ePAD in furazolidone solutions with different concentrations were measured. Figure 5b shows the response of the paper-based electrochemical sensor to the different concentrations of furazolidone. The results showed that the peak current response increased as the concentration of furazolidone increased, which maintained a linear relationship over a wide range (0.69 μM −100 μM). Figure 5c shows a linear relationship between the peak current density and the concentration of furazolidone. The linear equation was *j* = –14.24*c*–2170.91 (R^2^ = 0.99), where *j* and *c* are the peak current density and the concentration of furazolidone solution, respectively. The detection limit of the paper-based electrochemical sensor was 0.23 μM. The paper-based electrodes had good linear ranges, which proved that the preparation of paper-based electrodes was successful.

### 3.6. Determination of Furazolidone in Human Serum and Fish Sample 

The diluted human serum was used as actual samples, and the results showed that the paper-based electrode sensor had a good recovery rate, which proved that the PtNP/COF _TFPB−DHzDS_@rGO/ePAD sensor has the potential to detect furazolidone in real examples (Appendix A). In addition, fish were raised for three days and given different amounts of furazolidone in the water (Appendix A). The water in the tank was used for the actual sample testing, and the results are shown in Appendix A, which indicates that a part of furazolidone was absorbed by the fish.

## 4. Conclusions

In conclusion, an electrochemical sensing platform based on a homemade paper-based electrode loaded with PtNP/COF_TFPB−DHzDS_@rGO composite was developed to detect furazolidone. The rGO−NH_2_ was used to guide the growth of COF_TFPB−DHzDS_ on its surface to prepare COF_TFPB−DHzDS_@rGO composites in which COF_TFPB−DHzDS_ were covalently linked on rGO−NH_2_. Then, Pt^4+^ was first coordinated with the N and S atoms of COF_TFPB−DHzDS_, and subsequently, PtNP/COF_TFPB−DHzDS_@rGO was obtained by reducing Pt^4+^. COF_TFPB−DHzDS_ was uniformly distributed on the layered rGO, and the ultra-small PtNPs were formed on COF_TFPB−DHzDS_@rGO as nanozymes. The rGO and PtNPs increased the electrical conductivity of COF_TFPB−DHzDS_, and the catalytic activity of PtNP/COF_TFPB−DHzDS_@rGO was enhanced. Therefore, the proposed furazolidone electrochemical sensor based on PtNP/COF_TFPB−DHzDS_@rGO nanocomposites had a low detection limit (5 nM), a wide determination range (15 nM−110 μM) based on GCE, and good repeatability and stability. In contrast, the limit of detection for the paper-based electrode was 0.23 μM and the linear range was 0.69–100 μM. In addition, this work provides a reference for the covalent attachment of COFS materials on the surface of rGO and the synthesis of ultra-small-sized nanoparticles.

## Data Availability

The data are available under the request to the correspondence.

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
