# Peer review of "A Paper-Based Electrochemical Sensor Based on PtNP/COFTFPB−DHzDS@rGO for Sensitive Detection of Furazolidone"

_biosensors, 2022, doi:10.3390/bios12100904_

Round 1

Reviewer 1 Report

The authors reported a Paper-Based Electrochemical Sensor Based on PtNPs/COFTFPB−DHzDS@rGO for Sensitive Detection of Furazolidone. Some considerations are presented below.

* In item 2.5 Preparation of paper-based electrode (PBE), the authors should better describe the preparation. They could present a schematic or real images of the preparation of the PBE.

* Authors should present and discuss the electrochemical (redox) process of the antibiotic studied. What is the redox mechanism that occurs with the studied molecule?

 * In line 294, the authors describe “wide range (15 nM−110 μM).”, but in figure 5a-b another range of concentrations is shown (10 μM - 110 μM).

 * The authors could have performed and presented a real sample analysis in the present work.

Author Response

Thank you very much for giving us very valuable suggestions. We have considered your questions seriously, and answered them as below.

Reviewer 1:

The authors reported a Paper-Based Electrochemical Sensor Based on PtNPs/COFTFPB−DHzDS@rGO for Sensitive Detection of Furazolidone. Some considerations are presented below.

1.Q: In item 2.5 Preparation of paper-based electrode (PBE), the authors should better describe the preparation. They could present a schematic or real images of the preparation of the PBE.

R: Thank you very much for your relevant advice. According to your advice, we have provided a schematic of the fabrication of paper-based electrodes. (See Scheme 2 in the revised manuscript).

2.Q: Authors should present and discuss the electrochemical (redox) process of the antibiotic studied. What is the redox mechanism that occurs with the studied molecule?

R: Thank you very much for your relevant advice. According to your advice, the mechanism for quantitative monitoring of antibiotic has been revised as followed. “According to previous report [39], the redox mechanism is speculated to be that the nitrogroup contained in the structure of FUZ is reduced under the synergistic catalysis of PtNPs and rGO, which generates an electrochemical signal at −0.4 V. With the enrichment of the detected substance, the concentration increases, further the electrochemical signal increases. PtNPs can catalyze the reduction of FUZ, and rGO is beneficial to improve the conductivity. Under the synergistic effect of PtNPs and rGO, the peak current becomes larger. Due to the existence of PtNPs, the overpotential of the reaction is reduced, making FUZ easier to reduce. So, the more positive the potential is, the more the peak potential shifts to the right.” (See line 315 on page 9 in the revised manuscript).

3.Q: In line 294, the authors describe “wide range (15  nM−110  μM).”, but in figure 5a-b another range of concentrations is shown (10 μM - 110 μM).

R: Thank you very much for your relevant advice. The detection limit, LD, is the lowest amount of analyte in a sample that can be detected with a specified degree of certainty (M.A. Castillo, R.C. Castells, Journal of Chromatography A, 921 (2001) 121–133). LD depends on blank signal (µb1) and response dispersion at the blank level (σb1). The detection limit could be estimated according to the following equation (M.A. Castillo, R.C. Castells, Journal of Chromatography A, 921 (2001) 121–133):

LD= 3.29 (σb1/b1)                     (1)

where b1 is the slope of the calibration line.

The linear range is the interval of analyte amount over which the method behaves linearly. The quantitation limit (LQ) defines the lower end of the linear range; the upper end is usually imposed by instrumental factors. The quantitation limit could be estimated according to the following equation (M.A. Castillo, R.C. Castells, Journal of Chromatography A, 921 (2001) 121–133):

LQ= kQ (σb1/b1)                      (2)

IUPAC (L. A. Currie, Anal. Chim. Acta, 391 (1999) 105-126) proposes a default value kQ=10.

In this work, the σb1 was obtained for 10 successive measurements of µb1. The calibration line is y1 = –84.6511–3.2922x. The slope of the calibration line is –3.2922. In other words, the value of b1 is –3.2922. So, the detection limit is 5  nM. The quantitation limit (LQ) is 15  nM. The upper end of the calibration line is according to the curve. When the concentration is greater than 110  μM, the line seems to bend. So, the upper end of the calibration line is 110  μM. Therefore, the linear calibration range is 15  nM−110  μM. Figure 5b is DPV of PtNPs/ COFTFPB−DHzDS @rGO/GCE in N2 0.2 M PBS (pH=7) in the presence of furazolidone from 0.69  μM to 110  μM, which is not the linear range. (See revised Supporting Information)

4.Q: The authors could have performed and presented a real sample analysis in the present work.

R: Thank you very much for your relevant advice. We have provided the real sample analysis (human serum and fish) based on paper-based electrode (See Table S1 and Table S2 in Supporting Information).

Reviewer 2 Report

This confusing work reports the development of a paper-based electrochemical sensor based on a cluster-like covalent organic framework containing platinum nanoparticles and reduced graphene oxide for the determination of furazolidone. The text is sloppy and the manuscript is quite confusing. The authors describe more results using a modified GCE than the paper electrode. The development of the analytical method was carried out on the modified GCE and not on the paper electrode. In addition, preparing the paper electrode is quite confusing. It is not possible to understand if a paper electrode was prepared to contain the three printed electrodes, or if it was used only as a working electrode (Figure 6a). Therefore, I recommend that the manuscript be rejected.

General comments:

1. The manuscript was submitted to a “Paper-based biosensors” section. However, it does not contain any bio elements in its composition. In addition, more results are presented using a modified GCE than on the paper electrode.

2. The electrode preparation needs to be better explained. What was the cutting printer used? Does the printed electrode contain the three electrodes? Why was the reference electrode coated with silver powder if a conventional cell is shown in Figure 6a. Insert a schematic of how the paper electrodes were prepared. Insert the paper electrode design.

3. The authors, like many others, confuse the terms "detection" and "determination". Detection is qualitative by nature, while determination always is quantitative. Qualitative analysis is the detection of the presence of ions or compounds in an unknown sample, for example. The term "determination" refers to quantitative analysis to obtain data on the amount of analyte by weight or by concentration of an element or a compound in a sample. Therefore, most of the words “detection" in the manuscript should be replaced by the term "determination" (or "quantitation" or "assay") if quantitative assays are involved.

Specific comments:

1. Introduction:

a. Lines 61-63. Lines 58-60. Add "accuracy" and "portability of equipment" to the list of characteristics of electroanalytical methods. Add the Chemosensors, 10 (2022) 357 reference to validate this information.

2. Materials and methods:

a. Line 97. Do not use the acronym PBS for "phosphate-buffered solution". PBS is the accepted acronym for "phosphate-buffered saline" which contains 0.9% NaCl to warrant physiological ionic strength. See the Sigma Aldrich catalog (product P5368), for example. Also, see: https://en.wikipedia.org/wiki/Saline (medicine). Unfortunately, PBS is often wrongly used as an acronym for "phosphate buffer(ed) solution" in the literature but this is wrong and can cause confusion.  Does the buffer employed by the authors really contain 0.9% NaCl (or other electrolytes such as MgCl2)? If yes, please specify. See: https://en.wikipedia.org/wiki/Phosphate-buffered_saline.

b. Line 96. “0.2 M nitrogen-saturated phosphate”. Nitrogen? Check.

c. Line 98. The correct unit for ultrapure water resistivity is MW cm. 

d. Standardize the description of the reagents and equipment: model (company, country).

e. Provide the company and country of the hydrophobic paper and flexible graphite paper.

3. Results and discussion:

a. The subsections in the Results and Discussion section are messed up. There are two subsections 3.1. The position of figures and text is also confusing.

b. Line 177. “3.1. Characterization of Characterization”. The text is sloppy.

c. The modified paper electrode needs to be characterized by CV and EIS. Not the GCE.

d. “Generally speaking, compared with the three-electrode system composed of glassy carbon electrodes, the paper-based electrodes have similar experimental results, which proves that the preparation of paper-based electrodes is successful”. The LOD obtained with the modified GCE was 5.0 nM, while the paper electrode was 0.23 μM. Nothing similar. Very distant values.

4. Conclusion:

a. “Therefore, the proposed furazolidone electrochemical sensor based on PtNPs/ COFTFPB−DHzDS @rGO nanocomposites had a low detection limit (5 nM)”. The LOD value of 5.0 nm was obtained using the MODIFIED GCE, not the paper-based sensor that was supposed to be the subject of the manuscript.

Author Response

Thank you very much for giving us very valuable suggestions. We have considered your questions seriously, and answered them as below.

Reviewer 2:

1.Q: The manuscript was submitted to a “Paper-based biosensors” section. However, it does not contain any bio elements in its composition. In addition, more results are presented using a modified GCE than on the paper electrode.

R: Thank you very much for your relevant advice. First, furazolidone is a nitro-containing antibiotic drug, which has been widely used in aquaculture and veterinary medicine to minimize the acute effects of Escherichia coli, Salmonella and other infections. Residues of antibiotics are found in aquatic products and all kinds of animals, owing to the accumulation of furazolidone through the food chain, which will be detrimental to the human body's immune system. Therefore, furazolidone is relevant to biology. To prove that this work is biologically relevant, we measured furazolidone in serum and fish. The results indicated that part of furazolidone was absorbed by fish. Second, the main purpose of this work is to realize the sensitive detection of furazolidone based on portable paper-based electrodes. Since the traditional GCE was extensively used in lab, the sensitive detection of furazolidone was firstly carried out on modified GCE to explore the feasibility of the idea. Then the detection of furazolidone was carried out by using the as-prepared paper-based electrodes. The experimental results show that the paper-based electrode has good linearity, so the paper-based electrode is expected to be popular and replace the expensive traditional electrode. (See the Supporting Information)

2.Q: The electrode preparation needs to be better explained. What was the cutting printer used? Does the printed electrode contain the three electrodes? Why was the reference electrode coated with silver powder if a conventional cell is shown in Figure 6a. Insert a schematic of how the paper electrodes were prepared. Insert the paper electrode design.

R: Thank you very much for your relevant advice. First, the commercial carbon paper was cut into long strips with 3 mm in width and 3 cm in length by using a regular paper knife. Then two sides of a white cardboard with 2 cm in width and 3 cm in length were painted by white nail polish. Next three long carbon paper strips were pasted onto a side of the white cardboard with an interval of about 0.5 cm. The middle section of long carbon paper strips (about one-thirds of long strips) were painted with white nail polish. The bottom section of long carbon paper strips was peeled off by acrylic transparent tape to get new surface with graphite-like foam as electrodes (working electrode, reference electrode and counter electrode). Then, the reference electrode was coated with a layer of conductive silver powder, and HCl was added dropwise to form AgCl/Ag. A schematic fabrication of paper-based electrodes was illustrated in Scheme 2 in the revised manuscript. (See Scheme 2 in the revised manuscript).

3.Q: The authors, like many others, confuse the terms "detection" and "determination". Detection is qualitative by nature, while determination always is quantitative. Qualitative analysis is the detection of the presence of ions or compounds in an unknown sample, for example. The term "determination" refers to quantitative analysis to obtain data on the amount of analyte by weight or by concentration of an element or a compound in a sample. Therefore, most of the words “detection" in the manuscript should be replaced by the term "determination" (or "quantitation" or "assay") if quantitative assays are involved.

R: Thank you very much for your relevant advice. According to your suggestion, The words “detection" in the manuscript had been changed to "determination" in the whole article. (See the revised manuscript).

4.Q: 1. Introduction: a. Lines 61-63. Lines 58-60. Add "accuracy" and "portability of equipment" to the list of characteristics of electroanalytical methods. Add the Chemosensors, 10 (2022) 357 reference to validate this information.

R: Thank you very much for your relevant advice. The “accuracy" and "portability of equipment” has been added. In addition, the Chemosensors, 10(2022) 357 has also been cited in the revised manuscript. (See the revised manuscript).

5.Q: 2. Materials and methods:

  1. Line 97. Do not use the acronym PBS for "phosphate-buffered solution". PBS is the accepted acronym for "phosphate-buffered saline" which contains 0.9% NaCl to warrant physiological ionic strength. See the Sigma Aldrich catalog (product P5368), for example. Also, see: https://en.wikipedia.org/wiki/Saline (medicine). Unfortunately, PBS is often wrongly used as an acronym for "phosphate buffer(ed) solution" in the literature but this is wrong and can cause confusion. Does the buffer employed by the authors really contain 0.9% NaCl (or other electrolytes such as MgCl2)? If yes, please specify.

See: https://en.wikipedia.org/wiki/Phosphate-buffered saline.

R: Thank you very much for your relevant advice. The “PBS” has been changed to “phosphate buffer solution” in the whole article. (See the revised manuscript).

  1. Line 96. “0.2 M nitrogen-saturated phosphate”. Nitrogen? Check.

R: Thank you very much for your relevant advice. The “0.2 M nitrogen-saturated phosphate” has been changed to “0.2 M N2-saturated phosphate buffer solution”. (See the revised manuscript).

  1. Line 98. The correct unit for ultrapure water resistivity is MΩ cm.

R: Thank you very much for your relevant advice. The “M ω cm” has been changed to “MΩ cm”. (See the revised manuscript).

  1. Standardize the description of the reagents and equipment: model (company, country).

R: Thank you very much for your relevant advice. The description of the reagents and equipment had been standardized. (See the revised manuscript).

  1. Provide the company and country of the hydrophobic paper and flexible graphite paper.

R: Thank you very much for your relevant advice. The company of “flexible graphite paper” has been provided. Flexible graphite paper was purchased from Jinglong Special Carbon Technology Co., Ltd (Beijing, China). The white cardboard was photopaper was purchased from the local market. (See the revised manuscript).

  1. Results and discussion:
  2. The subsections in the Results and Discussion section are messed up. There are two subsections. The position of figures and text is also confusing.

R: Thank you very much for your relevant advice. The “3.1. Characterization of PtNPs/COFTFPB−DHzDS@rGO” has been changed to “3.2. Characterization of PtNPs/COFTFPB−DHzDS@rGO”. In addition, the position of the image has also been rearranged. (See the revised manuscript).

  1. Line 177. “3.1. Characterization of Characterization”. The text is sloppy.

R: Thank you very much for your relevant advice. The “3.1. Characterization of Characterization” has been changed to “3.1. Characterization of PtNPs/COFTFPB−DHzDS@rGO”. (See the revised manuscript).

  1. The modified paper electrode needs to be characterized by CV and EIS. Not the GCE.

R: Thank you very much for your relevant advice. The CV and EIS of paper-based electrode had been provided (See the revised manuscript).

  1. “Generally speaking, compared with the three-electrode system composed of glassy carbon electrodes, the paper-based electrodes have similar experimental results, which proves that the preparation of paper-based electrodes is successful”. The LOD obtained with the modified GCE was 5.0 nM, while the paper electrode was 0.23 μM. Nothing similar. Very distant values.

R: Thank you very much for your relevant advice. The“Generally speaking, compared with the three-electrode system composed of glassy carbon electrodes, the paper-based electrodes have similar experimental results, which proves that the preparation of paper-based electrodes is successful”. The LOD obtained with the modified GCE was 5.0 nM, while the paper electrode was 0.23 μM” has been changed to “Generally speaking, compared with the three-electrode system composed of glassy carbon electrodes, the paper-based electrodes have good linear range, which proves that the preparation of paper-based electrodes is successful.” (See the revised manuscript).

Reviewer 3 Report

The manuscript “A Paper-Based Electrochemical Sensor Based on PtNPs/COFTFPB−DHzDS@rGO for Sensitive Detection of Furazolidone” developed an paper-based electrochemical biosensor for the detection of Furazolidone. The authors demonstrated meticulous characterizations of such system (morphology, electrochemical behaviors, selectivity, repeatability, and stability) with conventional electrodes. They also demonstrate a practical utility of the system, using paper-based electrodes.

The work is interesting and precedent, although it is preliminary.

I think it is acceptable after some revision, taking into account the following points.

Major points:

1.    In the Abstract, the author need to explicitly state the limit of detection and linear range for paper-based electrode. Stating “paper-based electrode loaded with PtNPs/ COFTFPB−DHzDS @rGO had similar results. “(Line 22) is not clear enough to the audience the exact results of paper-based electrode.

2.    In the Conclusion, line 353, the limit of detection is stated as 5 nM. From section 3.5, we could know that 5 nM seems to be the LOD of conventional electrodes. Once again, the authors need to mention the performance of paper-based electrode.

Minor points:

3.    Line 177. “3.1. Characterization of Characterization of PtNPs/COFTFPB−DHzDS@rGO” should be 3.2.

4.    Figure 2. (f). the inserted figure is hard to read. It is okay to either have inserted figure as a stand-alone figure or have it in the Supplementary Information (SI).

5.    Sample size is missing from data presentation. Figure 4, 5, 6  should fully describe data presentation with sufficient details (e.g., Mean ± SD). What is the sample size in Figures?

6.    Figure 5. (d) Text font is too small to read. In the paragraph below (Line 308), the substrate tested in the selectivity test is not listed, so that it is harder for the audience to appreciate the selectivity of your system.

Author Response

Thank you very much for giving us very valuable suggestions. We have considered your questions seriously, and answered them as below.

Reviewer 3:

1.Q: In the Abstract, the author need to explicitly state the limit of detection and linear range for paper-based electrode. Stating “paper-based electrode loaded with PtNPs/ COFTFPB−DHzDS @rGO had similar results. “(Line 22) is not clear enough to the audience the exact results of paper-based electrode.

R: Thank you very much for your relevant advice. According to your advice, we have provided the limit of detection and linear range for paper-based electrode as followed. “In contrast, the detection limit for the paper-based electrode is 0.23 μM and the linear range is 0.69 μM -110 μM.” (See line 21 on page 1 in the revised manuscript).

2.Q: In the Conclusion, line 353, the limit of detection is stated as 5 nM. From section 3.5, we could know that 5 nM seems to be the LOD of conventional electrodes. Once again, the authors need to mention the performance of paper-based electrode.

R: Thank you very much for your relevant advice. According to your advice, we have provided the limit of detection and linear range for paper-based electrode in conclusion. “In contrast, the detection limit for the paper-based electrode is 0.23 μM and the linear range is 0.69 μM -110 μM.” (See line 378 on page 11 in the revised manuscript).

3.Q: Line 177. “3.1. Characterization of Characterization of PtNPs/COFTFPB−DHzDS@rGO” should be 3.2.

R: Thank you very much for your relevant advice. According to your advice, the “3.1. Characterization of Characterization of PtNPs/COFTFPB−DHzDS@rGO” has been changed to “3.2. Characterization of PtNPs/COFTFPB−DHzDS@rGO”. (See the revised manuscript)

4.Q: Figure 2 (f). the inserted figure is hard to read. It is okay to either have inserted figure as a stand-alone figure or have it in the Supplementary Information (SI).

R: Thank you very much for your relevant advice. According to your advice, the supporting information has been provided, the inset in Figure 2f has been moved to Supporting Information. (See the Supporting Information)

5.Q: Sample size is missing from data presentation. Figure 4, 5, 6 should fully describe data presentation with sufficient details (e.g., Mean ± SD). What is the sample size in Figures?

R: Thank you very much for your relevant advice. In figure 4, 5, 6, the data plotted for the calibration curve are the average of the five experiments, the sample size has been described and the length of the error bar indicates the magnitude of the relative deviation. (See the revised manuscript )

6.Q: Figure 5. (d) Text font is too small to read. In the paragraph below (Line 308), the substrate selectivity of your system.

R: Thank you very much for your relevant advice. According to your advice, the text font was enlarged in Figure 4d. In addition, the substrate (glucose, sodium carbonate, thiourea, NaCl, UA, DA, mannose) tested in the selectivity test had been listed. (See Figure 4d in the revised manuscript)

Round 2

Reviewer 2 Report

The sloppy manuscript describes the development of a paper-based electrochemical sensor based on PtNPs/COFTFPB−DHzDS@rGO for sensitive determination of furazolidone. The performance of the paper-based electrode is compared to a GCE modified with the same modifying agents. After Revision 1, the manuscript has been improved. However, there are still some questions to be resolved and improvements needed. Therefore, I recommend that a major review be carried out in order for the manuscript to be published in Biosensors.

1. The authors do not understand what a biosensor is. According to IUPAC, a biosensor is an electrochemical sensor that has a biological recognition element. Examples: Sensor having a type of chemically-modified electrode or ion-selective electrode modified with enzyme, antigen/antibody, certain Langmuir-Blodgett films, liposomes, plant or animal tissue, DNA, etc. See information in Pure and Applied Chemistry, 92 (2020) 641–694. It's okay to develop just modified electrodes. My questioning was only due to the special section of the journal that the manuscript was developed for. And after the wrong answer of the authors about biosensors, it was necessary to make this clarification.

2. Line 97. The correct unit for ultrapure water resistivity is MΩ cm. I believe there was a misconfiguration from Ω to W.

3. Lines 63-67. The introduction needs to be improved with more information on paper-based electrodes. So, add the following sentence: “Electrochemical paper-based analytical devices usually consist of a three-electrode setup integrated into a paper substrate, offering several benefits, such as reduced consumption of reagents and samples, portability, low cost, and availability of the raw material”. Add the reference Microchemical Journal, 179 (2022) 107588 to validate this information.

4. The results obtained with the paper electrode should also be entered in Table 1.

5. The authors did not understand the present reviewer's questioning about the terms "detection” and “determination”, and thus, they messed up the manuscript. The substitution of detection for determination was only for excerpts referring to quantification. LOD is the limit of detection, not of determination. There is no term "limit of determination". Please carefully review the manuscript.

6. One more mess. "The linear equation was ip" ... where ip and c were the peak current ... (Lines 322-323). "The linear equation was ip... where ip and c were the peak current density... (Lines 351-352). However, the graphs are a function of j (current density).

7. Do not abbreviate paper-based electrode as PBE. Use the acronym ePAD, which is commonly used in the literature. ePAD is an electrochemical paper-based device. Use ePAD.

8. Looking at the voltammograms and Nyquist spectra in Figure 5, it is seen that the best results were observed using the bare paper electrode (curve a) and not using the modified paper electrode PtNPs/COFTFPB−DHzDS@rGO/PBE (curve b). What a mess.

9. What does "EC" mean in the voltammograms in Figure 5? Was the reference electrode on the paper electrodes not Ag/AgCl? A general confusion.

10. It is necessary to add a sample preparation section in the experimental section. Inform where (city and country) the samples were obtained. What specie of fish is used? As a live animal was used, a statement from the University's Ethics Board is required.

Author Response

Thank you very much for giving us very valuable suggestions. We have considered your questions seriously, and answered them as below.

1.Q: The authors do not understand what a biosensor is. According to IUPAC, a biosensor is an electrochemical sensor that has a biological recognition element. Examples: Sensor having a type of chemically-modified electrode or ion-selective electrode modified with enzyme, antigen/antibody, certain Langmuir-Blodgett films, liposomes, plant or animal tissue, DNA, etc. See information in Pure and Applied Chemistry, 92 (2020) 641–694. It's okay to develop just modified electrodes. My questioning was only due to the special section of the journal that the manuscript was developed for. And after the wrong answer of the authors about biosensors, it was necessary to make this clarification.

R: Thank you very much for your relevant advice. We deeply apologize for the misunderstanding of the biosensor. We didn't do experiments related to enzymes, or antigens and antibodies. Enzymes are highly efficient biocatalysts with high activity and selectivity, which play an increasingly important role in pharmaceutical, chemical, food processing and other industrial production. However, the high cost of enzymes and their fragility often make it difficult to maintain good conformation and high catalytic activity under complex industrial operations and harsh application conditions.

In order to overcome the shortcomings of the enzymes mentioned above, nanomaterial with lenzyme-like catalytic activity, which are called as nanozyme, are developed to replace enzymes for biosensor construction. The low economic cost, high stability and multifunctional physicochemical properties of nanozymes accurately overcome the inherent instability and high cost of enzymes, while offering more potential for rational design and functionalization. In our work, the PtNPs/COFTFPB−DHzDS@rGO nanocomposites were prepared for sensitive detection of furazolidone, in which PtNPs as nanozymes can play a catalytic role. In addition, we investigated in vivo related experiments with biologically inspired components that are accorded with the requirements of the magazine. Therefore, we sincerely hope that the manuscript will be a good fit for the magazine.

2.Q: Line 97. The correct unit for ultrapure water resistivity is MΩ cm. I believe there was a misconfiguration from Ω to W.

R: Thank you very much for your relevant advice. the “M ω cm” has been changed to “MΩ cm” (See the revised manuscript).

3.Q: Lines 63-67. The introduction needs to be improved with more information on paper-based electrodes. So, add the following sentence: “Electrochemical paper-based analytical devices usually consist of a three-electrode setup integrated into a paper substrate, offering several benefits, such as reduced consumption of reagents and samples, portability, low cost, and availability of the raw material”. Add the reference Microchemical Journal, 179 (2022) 107588 to validate this information.

R: Thank you very much for your relevant advice. We have added the following sentence: “Electrochemical paper-based analytical devices usually consist of a three-electrode setup integrated into a paper substrate, offering several benefits, such as reduced consumption of reagents and samples, portability, low cost, and availability of the raw material” in the introduction. In addition, the reference Microchemical Journal, 179 (2022) 107588 also was added in the part of reference.”

4.Q: The results obtained with the paper electrode should also be entered in Table 1.

R: Thank you very much for your relevant advice. The results obtained with the paper-based electrode have been added in Table 1 (See the revised manuscript).

5.Q: The authors did not understand the present reviewer's questioning about the terms "detection” and “determination”, and thus, they messed up the manuscript. The substitution of detection for determination was only for excerpts referring to quantification. LOD is the limit of detection, not of determination. There is no term "limit of determination". Please carefully review the manuscript.

R: Thank you very much for your relevant advice. I'm so sorry for misunderstanding the concepts of determination and detection, so I revised the manuscript again according to your advice. The determination was used in quantitative part, I, while the detection was used in the part of qualitative. In addition, the "limit of determination" has been changed into “limit of detection” (See the revised manuscript).

6.Q: One more mess. "The linear equation was ip" ... where ip and c were the peak current ... (Lines 322-323). "The linear equation was ip... where ip and c were the peak current density... (Lines 351-352). However, the graphs are a function of j (current density).

R: Thank you very much for your relevant advice. The experimental data was plotted according to the current density. Due to our carelessness, ip was wrongly expressed. I am deeply sorry. This misstatement has been corrected. “where ip and c were the peak current density …” has been changed into “where jp and c were the peak current density” (Lines 322-323).

7.Q: Do not abbreviate paper-based electrode as PBE. Use the acronym ePAD, which is commonly used in the literature. ePAD is an electrochemical paper-based device. Use ePAD.

R: Thank you very much for your relevant advice. According to your advice, the paper-based electrode was abbreviated into “ePAD” in the whole article (See the revised manuscript).

8.Q: Looking at the voltammograms and Nyquist spectra in Figure 5, it is seen that the best results were observed using the bare paper electrode (curve a) and not using the modified paper electrode PtNPs/COFTFPB−DHzDS@rGO/PBE (curve b). What a mess.

R: Thank you very much for your relevant advice. We would like to explain that the results of this experiments. Firstly, CV and EIS use Fe (CN)63−/4− as a signal probe, and Fe (CN)63−/4− is negatively charged. Although PtNPs and rGO in the target material are conducive to improving the conductivity, the conductivity of COF is generally not good. If the COF is expected to be negatively charged or uncharged, the target material may still not conducive to electron transfer. Therefore, a slight decrease in the corresponding peak current and an increase in the impedance value are normal. The same experimental results were also obtained based on GCE, Fig. 3a-b, which was explained in line 223-232. Secondly, the purpose of modifying the target material is to catalyze the FUZ, so as to generate corresponding electrochemical signals. If there is no modification of the target material, there will be no electrochemical signals in DPV (Fig. 4a).

9.Q: What does "EC" mean in the voltammograms in Figure 5? Was the reference electrode on the paper electrodes not Ag/AgCl? A general confusion.

R: Thank you very much for your relevant advice. The “EC” has been changed into Ag/AgCl. (See the revised manuscript).

10.Q: It is necessary to add a sample preparation section in the experimental section. Inform where (city and country) the samples were obtained. What specie of fish is used? As a live animal was used, a statement from the University's Ethics Board is required.

R: Thank you very much for your relevant advice. We have provided the preparation of the sample in the SI as followed. “The goldfish was purchased from Nanchang (Jiangxi, China). The fish were raised in 1 L of water, and different dose of furazolidone was given daily. After 3 day, the water was configured as real samples.” The “Compliance with ethical standards: All experiments involving animals were in accordance with the guidelines of the National Institute of Food and Drug, Nanchang, China, and approved by the institutional ethical committee (IEC) of Jiangxi University of Traditional Chinese Medicine. This article does not contain any studies with human participants performed by any of the authors.” has been added in the revised manuscript.